# Machine Learning White-Hat Worm Launcher for Tactical Response by Zoning in Botnet Defense System [note 1]

**DOI:** 10.3390/s22134666

**Published:** 2022-06-21

**Authors:** Xiangnan Pan, Shingo Yamaguchi

**Affiliations:** Graduate School of Sciences and Technology for Innovation, Yamaguchi University, Ube 755-8611, Japan; pan25010@gmail.com

**Keywords:** botnet, BDS, machine learning (ML), zoning, IoT, white-hat, Petri net

## Abstract

Malicious botnets such as Mirai are a major threat to IoT networks regarding cyber security. The Botnet Defense System (BDS) is a network security system based on the concept of “fight fire with fire”, and it uses white-hat botnets to fight against malicious botnets. However, the existing white-hat Worm Launcher of the BDS decides the number of white-hat worms, but it does not consider the white-hat worms’ placement. This paper proposes a novel machine learning (ML)-based white-hat Worm Launcher for tactical response by zoning in the BDS. The concept of zoning is introduced to grasp the malicious botnet spread with bias over the IoT network. This enables the Launcher to divide the network into zones and make tactical responses for each zone. Three tactics for tactical responses for each zone are also proposed. Then, the BDS with the Launcher is modeled by using agent-oriented Petri nets, and the effect of the proposed Launcher is evaluated. The result shows that the proposed Launcher can reduce the number of infected IoT devices by about 30%.

## 1. Introduction

The Internet of Things (IoT) has become an exciting technology in the world, and it has grown into the largest platform of millions of IoT devices interacting with each other to improve our daily lives. However, such a popular IoT can also be used as a potential tool for uploading malicious entities. These entities use IoT vulnerabilities caused by various constraints such as limited resources and weak security to access legitimate devices and can further take various forms of attack [1]. For example, distributed denial of service (DDoS) in IoT networks is a very popular malicious attack, which disrupts the normal traffic of the target server by launching multiple consecutive requests from distributed IoT devices, eventually causing the server to crash [2,3,4].

Mirai, a malware capable of launching DDoS attacks, is also accountable for the largest record of attack traffic to date, targeting a French-managed service provider and up to 15 million IoT devices at a maximum speed of 1 Tbps [5]. The most remarkable thing is that the source code of Mirai was published by its author [6]. This may give chances for unethical behavior on the Internet. So far, the devices infected by Mirai have already reached 164 countries; the speed of spread is amazing. It is estimated that the number of deployed IoT devices will be about 20–30 billion by 2030 [7].

Faced with such a huge number of IoT devices, one method is to use a kind of white-hat worm, such as Hajime, to fight Mirai [8]. The white-hat worm has two attributes, the lifespan and the secondary infection possibility [9]. These two attributes ensure that the white-hat worms can destroy themselves after the Mirai bots are eliminated by them. Yamaguchi [10] proposed a promising network security system called the Botnet Defense System (BDS). It can use white-hat botnets to eliminate malicious botnets like Mirai. Yamaguchi also proposed three basic strategies for white-hat worms according to the network situation and confirmed the effect of the BDS through an agent-oriented Petri net model (PN${}^{2}$ model) [11].

The BDS only considers the number of white-hat worms; there is no discussion on the position of white-hat worms. Pan et al. [12] proposed a machine-learning (ML)-based white-hat Worm Launcher for the BDS. In this ML-based Launcher, the given network is rigidly divided into several subnetworks with the divide-and-conquer algorithm (D&C algorithm) to adapt the machine learning models, and then, the white-hat worms’ positions are predicted by the ML models. The ML-based Launcher also has a good adaptability because of the D&C algorithm; even if the given network is very large, we can also successfully input the modified network into the ML models through the preprocessing step of “padding” or “multiple divide-and-conquer”. However, the Mirai botnet is not always spread uniformly over the network. Some parts have a high density of Mirai bots, while others have a low one. The D&C algorithm always divides the whole network into the same subnetworks and processes them by ML models, which will increase the time and resources. In general, there is some bias in the distribution of the Mirai bots, so the BDS should deal with the Mirai botnet reasonably according to the distribution, and it is necessary to optimize our ML white-hat worm Launcher.

In this paper, we propose a novel machine learning white-hat Worm Launcher for tactical response by zoning in the BDS. We introduce the concept of zoning to grasp the malicious botnet spread with bias over the IoT network. This enables the BDS to divide the network into areas called zones and make tactical responses for each zone. We also propose tactics for response to each zone. Then, we model the BDS with the Launcher by using agent-oriented Petri nets and evaluate the effect of the proposed Launcher. We use Mirai as a representative of malicious botnets. The proposed method is available to the other malicious botnets.

Our major contributions are as follows:(1)The concept of zoning is introduced to grasp malicious botnets’ spread with bias over IoT networks.(2)Tactical response by zone is proposed to build and operate white-hat botnets efficiently and effectively against the malicious botnets.(3)The effect of our white-hat Worm Launcher for BDS is confirmed through simulation evaluation with agent-oriented Petri nets. The result show that the proposed Launcher can reduce the number of infected IoT devices by about 30%.

The rest of this paper is organized as follows. Section 2 introduces the previous work and the BDS. Section 3 describes the zoning and the tactics to respond by zoning. Section 4 describes the simulation evaluation. Section 5 summarizes the results obtained in this paper and gives future work.

## 2. Related Work

### 2.1. Previous Work

Recent related studies on botnets can be roughly divided into two categories according to their purposes: detection and mitigation.

With the popularity of artificial intelligence (AI), the detection mechanisms of malicious network attacks are keen to use AI technology [13]. Nagisetty and Gupta [14] used four different deep learning models to construct a framework in the IoT backbone network to identify malicious network traffic. Soe et al. [15] proposed a high-performance attack detection system with three different machine learning (ML) algorithm sequential classifiers. Gupta et al. [16] proposed a deep-reinforcement-learning (DRL)-based detection framework with a dataset that contains IoT traces. Qu et al. [17] proposed an evaluation method based on percolation theory to quantify the security risk in a power CPS network. Qu et al. [18] also proposed a hybrid detection method with improved AKF and GRU-CNN to detect false data injection attacks. Wang et al. [19] proposed a new method to automatically mine attack patterns from measurement data and information alarm logs. Vishwakarma et al. [20] proposed a honeypot-based framework with machine learning techniques, which shows a high performance on zero-day DDoS attacks. In addition, the detection model usually needs to rely on reliable datasets, and some researchers also pay attention to the generation of network datasets or simulators. Guerra-Manzanares et al. [21] proposed an IoT dataset for ML-based detection frameworks, which includes actual malicious network traffic in a medium-sized IoT. Adan et al. [22] designed an Internet of Things attack simulator, which shows how Mirai works in DDoS attacks. Other botnet detection frameworks also include a blockchain-based mechanism [23,24,25] and an SDN-based mechanism [26].

Botnet mitigation measures are also very important. This aims to mitigate the threats from the DDoS. Gopal et al. [27] thought of a white-list-based solution to prevent the spread of Mirai and confirmed the blocking function against botnets through experiments. Manso et al. [28] proposed an intrusion detection system through a software layer, and it can launch a software-defined network controller to control the traffic when the system detects an attack. This can reduce the traffic and makes it possible to maintain normal traffic when an attack happens. Ceron et al. [29] proposed a method to handle the network traffic caused by IoT malware. The method can characterize the malware behaviors and modify the traffic through the network layer to limit malicious activities.

We briefly summarize the attributes and limitations of the above detection and mitigation methods in Table 1 and Table 2. The previous studies only focused on the detection methods or mitigation measures; they are not able to exterminate existing botnets.

### 2.2. Botnet Defense System

The BDS is a network security system based on the concept of “fight fire with fire”. It uses white-hat botnets to fight against malicious botnets like Mirai. Figure 1 illustrates the configuration of the BDS and an IoT system defended by the BDS. The upper part represents the structure of the BDS. It consists of four parts: Monitor, Strategy Planner, Worm Launcher, and C&C Server. The lower part represents the IoT system. The inner graph represents the network of the system. Each vertex represents a network node, and each node has a single device. Each edge represents a connection between network nodes. A black circle (
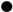
) and a white circle (
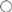
) respectively represent a malicious bot and a white-hat bot. When the Monitor detects a malicious botnet, the Worm Launcher will send the white-hat worm according to the strategy specified by the Strategy Planner. Yamaguchi also put forward three basic strategies. These strategies can adjust the number of white-hat worms according to not only the ability of the worms, but also the structure of the network. However, there is no discussion on the placement of white-hat worms; it should be reasonably considered and configured.

Pan et al. [12] proposed a machine learning white-hat Worm Launcher for the BDS. It basically has two phases: decision and placement. The first phase uses a feedforward neural network (FNN) to decide an appropriate placement tactic. The second phase uses a multitask learning model to predict white-hat worms’ appropriate position based on the predicted tactic. Furthermore, this Launcher takes a divide-and-conquer approach so that it can be adapted to large-scale IoT networks. The procedure of the Launcher is given as Procedure 1. The Launcher divides a given network into subnetworks. For each subnetwork, it predicts a tactic and, then, the white-hat worms’ placement based on the tactic.
**Procedure 1 **Worm-Launcher(N,DMirai,M1,M2).**Input:** Mirai bots’ distribution DMirai in network *N*, machine learning model M1 to decide tactics, machine learning model M2 to place white-hat worms**Output:** White-hat worms’ placement DW for DMirai  ▷ *Step 1: Divide Stage*  1: Divide *N* into *n* subnetworks N1, N2, ⋯, Nn by network scale  ▷ *Step 2: Conquer Stage*  2: **for** each subnetwork Ni in N1, N2, ⋯, Nn **do**  3:      Use M1 to decide a tactic Ti for Ni  4:      Use M2 to make white-hat worms’ placement DWi on Ni based on Ti  5: **end for**   ▷ *Step 3: Combining Stage*  6: Combine DW1, DW2, ⋯, DWn with the whole placement DW  7: Output DW and stop.


We illustrate the procedure with an example shown in Figure 2. Twenty devices are infected by Mirai in a 10 × 10 grid network. As a preparation, we trained two machine learning models, i.e., the FNN model $M_{1}$ and the multitask learning model $M_{2}$, for a small network. The small network actually corresponds to the subnetwork divided in the procedure:Step 1 divides the network into subnetworks by the network scale. Assume *n* = 4 in this example. That is, the 10 × 10 network is divided into four 25 × 25 subnetworks.Step 2 predicts a tactic with $M_{1}$ and, then, the white-hat worms’ placement based on the tactic with $M_{2}$ for each subnetwork.Step 3 combines the predictions of all subnetworks into the final white-hat worm’s placement.

## 3. White-Hat Worm Launcher for Tactical Response by Zoning

### 3.1. Zoning

The Mirai botnet is not always spread uniformly over the network. There may be some bias in the distribution of the Mirai bots. The BDS should deal with the Mirai botnet based on its bots’ distribution. We define a density of bots in some area of the network as follows.

**Definition** **1.**
*(Bots density) Given a distribution of bots on a network, for a specific area of the network, the density δ of the bots is given as follows:*

$$\delta \overset{\mathrm{def}}{=}\frac{Number\;of\;bots}{Number\;of\;devices\;in\;the\;area}\left[\%\right].$$



We introduce the concept of zoning to the BDS. The BDS divides a given IoT network into some zones and plans a tactic for each zone. Note that the zoning depends not on the network scale, but on the Mirai bots’ density. We can expect the BDS to predict a tactic appropriate for the bots’ density in each zone and to build a white-hat botnet effectively and efficiently.

For zoning, we used a density-based clustering algorithm called density-based spatial clustering of applications with noise (DBSCAN) [30]. Density-based clustering extracts areas that have a higher density than the rest as clusters. DBSCAN takes two parameters: ε and minPts. ε is the radius from a point. minPts is the minimum number of points needed to form a dense region. Given a set of points, DBSCAN classifies each point into three categories: Core, Border, and Noise. Core points are points with at least minPts neighbors within ε. Border points are points with no more than minPts neighbors within ε, but with Core points within ε. Noise points are points with no neighbors within ε. Finally, DBSCAN creates clusters from a collection of Core points and Border points. We formalize DBSCAN as follows.

**Definition** **1.**
*(DBSCAN) For a set P of points, a radius ε from a point, and the minimum number minPts of points needed to form a dense region, DBSCAN is given as a mapping:*

(1)
$$DBSCAN:\left(P,\varepsilon ,minPts\right)\mapsto C_{1},C_{2},\cdots ,C_{m},$$

*where $C_{1},C_{2},\cdots ,C_{m}$ are clusters of the points.*


The reason for using DBSCAN is that it can find non-linearly separable clusters unlike k-means or Gaussian mixture EM clustering. It can find even a cluster that is surrounded by others. This is very important for zoning. Another reason is it is easy to control since there are only two parameters.

Combining clusters with a similar density, we define an obtained area as a zone.

**Definition** **3**
*(Zone) Let C be any cluster obtained by DBSCAN and δ(C) be the bot density of C. For some range of bots density, a zone is the union of cluster C such that δ(C) is in the range.*


We used the following three zones in this paper:Danger Zone $Z_{Danger}$: contains a cluster *C* such that δ is $\underline{\delta_{Danger}}\leq \delta \left(C\right)\leq \bar{\delta_{Danger}}$ (=100%). Almost all devices in this zone are infected by Mirai.Safe Zone $Z_{Safe}$: contains a cluster *C* such that $\underline{\delta_{Safe}}$ (=0%) $\leq \delta \left(C\right)\leq \bar{\delta_{Safe}}$. Almost no devices in this zone are infected.Unknown Zone $Z_{Unknown}$: contains a cluster *C* such that δ is $\underline{\delta_{Safe}}<\delta \left(C\right)<\bar{\delta_{Danger}}$. The devices in this zone are at risk of being infected.

We propose a procedure for zoning, which divides the given network into the three zones. The procedure is given as Procedure 2. Simply put, we ran DBSCAN twice for zoning. The first is for finding the Danger Zone, where a point represents the position of an infected device. The second is for finding the Safe Zone, where we regard a point to represent the position of a normal (uninfected) device. The remaining areas are incorporated into the Unknown Zone.

**Procedure 2 **DBSCAN-based-Zoning (N,DMirai,Dnormal).**Input:** Mirai bots’ distribution DMirai in network *N*, normal devices’ distribution Dnormal on *N*.**Output:** Danger Zone ZDanger, Safe Zone ZSafe, Unknown Zone ZUnknown.    ▷ *Step 1: Finding Danger Zone*  1: PMirai←{the point of each Mirai bot DMiraiin the space ofN}.  2: C1Mirai,C2Mirai, ⋯, CmMirai←DBSCAN(PMirai,εMirai,minPtsMirai).  3: ZDanger←∅.  4: **for** each cluster *C* in C1Mirai,C2Mirai, ⋯, CmMirai **do**  5:    **if** δ(C)≥δDanger_ **then**  6:      ZDanger←ZDanger∪C.  7:    **end if**  8: **end for**    ▷ *Step 2: Finding Safe Zone*  9: Pnormal←{the point of each normal device inDnormalin the space ofN}.  10: C1normal,C2normal, ⋯, Cnnormal←DBSCAN(Pnormal,εnormal,minPtsnormal).  11: ZSafe←∅.  12: **for** each cluster *C* in C1normal,C2normal, ⋯, Cnnormal **do**  13:    **if** δ(C)≤δSafe¯ **then**  14:      ZSafe←ZSafe∪C.  15:    **end if**  16: **end for**    ▷ *Step 3: Finding Unknown Zone*  17: ZUnknown←(DMirai∪Dnormal)\(ZDanger∪ZSafe).    ▷ *Step 4*  18: Output ZDanger, ZSafe, ZUnknown and stop.


The DBSCAN algorithm is known to have difficulty determining the parameter values for a dataset with large density differences. To solve this problem, we decided to run the DBSCAN algorithm many times, depending on the density. In fact, Procedure DBSCAN-based-Zoning runs the DBSCAN algorithm twice.

We illustrate the proposed zoning procedure with an example shown in Figure 3. Figure 3a represents the Mirai bots’ distribution in a 10 × 10 grid network. In this example, we set $\varepsilon_{Mirai}=1$ and $minPts_{Mirai}=3$. This means that if Mirai has infected a device and it is surrounded by more than two neighbors, we regard them to be in danger because they form a cluster. Meanwhile, we set $\varepsilon_{normal}=1$ and $minPts_{normal}=5$. This means that if an uninfected device is surrounded by four uninfected devices, we regard them to be safe because there is no risk of infection. We can determine these values according to the security policy and network configuration. For example, if $\varepsilon_{Mirai}=1$ and $minPts_{Mirai}=2$, we identify Danger Zones with greater sensitivity. This may launch more white-hat bots than necessary. On the other hand, if $\varepsilon_{Mirai}=1$ and $minPts_{Mirai}=5$, we identify Danger Zones with lower sensitivity. This may build weak a white-hat botnet. There is a trade-off between the detection and the disinfection ability. These parameters provide adaptability to balance between them:Step 1 finds the Danger Zone. As a result of DBSCAN with $\varepsilon_{Mirai}=1$ and $minPts_{Mirai}=3$, we obtain two clusters $C_{1}^{Mirai}$ (red area in the upper right of Figure 3b) and $C_{2}^{Mirai}$ (red area in the lower right of Figure 3b). Since $\delta \left(C_{1}^{Mirai}\right)=\delta \left(C_{2}^{Mirai}\right)=100\%\geq \underline{\delta_{Danger}}$, these clusters comprise the Danger Zone, i.e., $Z_{Danger}=C_{1}^{Mirai}\cup C_{2}^{Mirai}$.Step 2 finds the Safe Zone. As a result of DBSCAN with $\varepsilon_{normal}=1$ and $minPts_{normal}=5$, we obtain one cluster $C_{1}^{normal}$ (blue area in the left of Figure 3b). For $\bar{\delta_{Safe}}=0$, since $\delta \left(C_{1}^{normal}\right)=0\%\leq \bar{\delta_{Safe}}$, the cluster becomes the Safe Zone, i.e., $Z_{Safe}=C_{1}^{Mirai}$.Step 3 finds the Unknown Zone. The remaining areas comprise $Z_{Unknown}$ (yellow areas of Figure 3b).Step 4 outputs $Z_{Danger}$, $Z_{Safe}$, $Z_{Unknown}$ as the Danger, Safe, and Unknown Zones.

### 3.2. Tactics by Zone

We propose tactics for sending white-hat worms by zone to build the white-hat botnet efficiently and effectively. The tactic for each zone is as follows.

#### 3.2.1. Surrounding Tactic $T_{Surrounding}$ for Danger Zone

In the Danger Zone, Mirai bots are clumped. The BDS should confine those Mirai bots within the zone. The surrounding tactic [31] places white-hat worms around the zone and prevents the Mirai bots from spreading. See Figure 4.

#### 3.2.2. Protecting Tactic $T_{Protecting}$ for Safe Zone

In the Safe Zone, there are few Mirai bots. Therefore, the BDS should prevent the Mirai botnet from entering this area. The protecting tactic [32] places white-hat worms on the border of the area to protect them like a firewall and prevent the bots from entering. See Figure 5.

#### 3.2.3. Machine Learning Tactic $T_{ML}$ for Unknown Zone

This tactic places white-hat worms according to the result of the machine learning models. See Figure 6. We used a multitask learning model to implement this tactic in the same way as the second phase of [12].

### 3.3. White-Hat Worm Launcher by Zoning

Combining the zoning procedure and the tactics for sending white-hat worms by zone, we propose a new white-hat Worm Launcher that can make tactical responses for the divided zones. The procedure of the Launcher is given as Procedure 3. The first step uses Procedure DBSCAN-based-Zoning to divide a given network into three zones, i.e., the Danger, Safe, and Unknown Zones. The second step makes the placement of white-hat worms according to the tactic for each zone, i.e., the surrounding tactic for the Danger Zone, the protecting tactic for the Safe Zone, and the machine learning tactic for the Unknown Zone. The third step combines those results for the whole placement.

We illustrate the proposed procedure with an example shown in Figure 7. The network and Mirai bots’ distribution are the same as the example of Figure 3:Step 1 divides the network into three zones by using Procedure DBSCAN-based-Zoning. As stated before, we obtain the result of zoning shown in Figure 3b.Step 2 places white-hat worms in each zone. Each zone takes a different tactic, i.e., the surrounding tactic for the Danger Zone, the protecting tactic for the Safe Zone, and the machine learning tactic for the Unknown Zone.Step 3 combines those results for the whole placement.
**Procedure 3** Zoning-based-Worm-Launcher(N,DMirai,Dnormal,M2).
**Input:** Mirai bots’ distribution DMirai in network *N*, normal devices’ distribution Dnormal on *N*, machine learning model to place white-hat worms M2.**Output:** White-hat worms’ placement DW for DMirai.    ▷ *Step 1: Zoning Stage*  1: ZDanger, ZSafe, ZUnknown←DBSCAN-based-Zoning(N,DMirai,Dnormal).    ▷ *Step 2: Placing Stage*  2: **for** each zone *Z* in ZDanger, ZSafe, ZUnknown **do**  3:     Place white-hat worms as follows:  4:     **if** *Z* is ZDanger **then**  5:      Use the surrounding tactic and store the placement result in DWDanger.  6:     **end if**  7:     **if** *Z* is ZSafe **then**  8:      Use the protecting tactic and store the placement result in DWSafe.  9:     **end if**  10:     **if** *Z* is ZUnknown **then**  11:      Use the machine Learning tactic and store the placement result predicted by M2 in DWUnknown.  12:     **end if**  13: **end for**    ▷ *Step 3: Combining Stage*  14: Combine DWDanger, DWSafe, DWUnknown with the whole placement DW.  15: Output DW and stop.


Procedure Zoning-based-Worm-Launcher can run in a few seconds on a PC with an Intel Core i7 2.6 GHz and 16 GB RAM. Zoning in Step 1 takes about one second. Step 2 decides the placement of white-hat bots for each zone. Zoning omits predictions by the learning model in some zones and allows for a significant reduction in computation time. As a result, it can run in one second. Combining in Step 3 hardly takes any time.

## 4. Simulation Evaluation

### 4.1. Experiment

We modeled the BDS with the proposed zoning-based Worm Launcher by using an agent-oriented Petri net PN${}^{2}$ and, then, carried out experiments to evaluate the effect of the Launcher through the simulation of the PN${}^{2}$ model. For the detail of the modeling by PN${}^{2}$, refer to [9].

We configured a 10 × 10 grid network *N* and considered four kinds of initial distributions of Mirai. The first three distributions and the result of zoning are illustrated in Figure 8:Distribution 1: Has a low density of Mirai bots in the whole network, i.e., δ(N)=31%. This is illustrated in Figure 8a. The infected network is divided into 25% Danger Zone, 50% Safe Zone, and 25% Unknown Zone.Distribution 2: Has a medium density of Mirai bots in the whole network, i.e., δ(N)=37%. This is illustrated in Figure 8b. The infected network is divided into 25% Danger Zone, 25% Safe Zone, and 50% Unknown Zone.Distribution 3: Has a high density of Mirai bots in the whole network, i.e., δ(N)=56%. This is illustrated in Figure 8c. The infected network is divided into 50% Danger Zone, 25% Safe Zone, and 25% Unknown Zone.Distribution 4: Set to be random. It has almost the same density as Distribution 1, i.e., δ(N)=30%; unlike Distribution 1, all the Mirai bots are placed randomly.

We used the tool PN2Simulator [33] for the simulation. We took Mirai’s infection rate $R_{Mirai}$ as a quantitative index. $R_{Mirai}$ is given by $R_{Mirai}={\#}_{Mirai}/{\#}_{device}$, where ${\#}_{Mirai}$ is the number of devices infected by Mirai and ${\#}_{device}$ is the number of all devices in *N*. The other simulation parameters are given as follows:Lifespan *ℓ* = 2;Delay time δ = 7 until rebooting;Secondary infection probability ρ = 50%.

### 4.2. Simulation Result

We compared the proposed Launcher with the Launcher of [12]. The notable difference between them having or not having zoning, i.e., the proposed Launcher has zoning, while the Launcher of [12] has no zoning. The simulation results can be seen in Table 3. Each value represents the mean of $R_{Mirai}$ at 1, 10, 100, 1000, and 10,000 steps for 1000 trials. In every distribution and step, the infection rate of the Launcher with zoning had to a lower value than the Launcher without zoning. Figure 9 charts these results to make it easier to compare these Launchers. From the graphs, we found that in the early stage, the Launcher with zoning removed significantly more Mirai bots than the Launcher without zoning. Especially in the case of Distribution 1, which contains 50% Safe Zone, the protecting tactic effectively prevented the devices in the zone from infection. However, the white-hat botnet behaves autonomously. Thus, the formation will collapse with the passage of time, and therefore, the effect of the initial launch position will disappear. This implies that the tactic has a time limit for its effect.

In addition, from the graph of Figure 9d, the random distribution resulted in a similar tendency as the other distributions. That is, the infection rate curve with zoning is steeper than without zoning, which means that the infection rate decreases more significantly. We confirmed that the proposed Launcher can also show performance in a general environment and effectively remove Mirai bots.

### 4.3. Discussion

Table 4 shows the positioning of this study among the related research. Most of the related research focuses on detecting botnets and mitigating the threat, while only the last three research works focus on exterminating existing botnets. The three research works deal with the BDS, but propose different Launchers. The Launcher of [10] gives only a basic functionality, which sends white-hat worms to the infected network. That is, it places worms at random and does not specify concrete nodes. On the other hand, the Launcher of [12] specifies nodes to launch. This Launcher learns how to place white-hat worms based on machine learning. The Launcher also divides the given network into small networks and increases the accuracy of the prediction. The Launcher proposed in this paper introduces the concept of zoning and applies tactics for each zone. In general, there is a bias in the distribution of Mirai bots. Therefore, the given network should be divided by the Mirai bots’ density rather than by network scale in a uniform manner. The proposed Launcher divides the given network into zones based on density and can use appropriate tactics depending on the density.

Next, we discuss tactical response by zoning. For the Safe and Danger Zones, we observed that the effect of the tactics for those zones was obvious, especially in the early stage, which showed that the white-hat botnet can effectively prevent the Mirai botnet from spreading as a “wall”. On the other hand, although it is not easy to isolate randomly distributed Mirai bots, our proposed Launcher showed excellent performance in the case of a random distribution. In general, the proposed Launcher can effectively reduce the Mirai infection rate and reduce the number of infected devices in the network. Zoning helps us make it easier to consider tactics because we separated the phenomenon. In addition, zoning saves time and cost during the design and runtime. This is important because machine learning often requires much computational time and cost.

The proposed Worm Launcher was designed to be scalable. In this paper, we used a 10 × 10 network to illustrate the proposed method. However, we can apply it to a network with a different size. At this time, we only have to perform padding or use the divide-and-conquer step given in our previous paper as the preprocess. This preprocess hardly takes any time. Actually, we performed a preliminary experiment using a larger network with 20 × 20 nodes and confirmed that the proposed Launcher is efficient. From the above, it can be said that the proposed Launcher is adaptable to large-scale networks.

## 5. Conclusions

In this paper, we proposed a novel machine learning white-hat Worm Launcher for tactical response by zoning in the Botnet Defense System. We first introduced the previous work and the BDS. Next, we proposed a new concept of zoning to grasp malicious botnets’ spread with bias over IoT networks. We also proposed three tactics to respond to each zone. Then, we confirmed the effect of the proposed white-hat Worm Launcher through a simulation evaluation with agent-oriented Petri net models. The result showed that the proposed Launcher is effective and can reduce the number of infected IoT devices by about 30% overall.

In this paper, we focused on the case of the Mirai botnet infecting only once. However, the Mirai botnet may infect many times. Whenever the Mirai botnet appears, we should nip it in the bud as early as possible. However, when a few Mirai bots are scattered, the proposed Launcher may not work well because all of the network is identified as a Safe Zone. To resolve this problem, we can use several approaches. For example, we can predict future infection situations and make a proactive response based on the prediction. We also can make a white-hat botnet resident to continually disinfect Mirai bots [34].

## Figures and Tables

**Figure 1 sensors-22-04666-f001:**
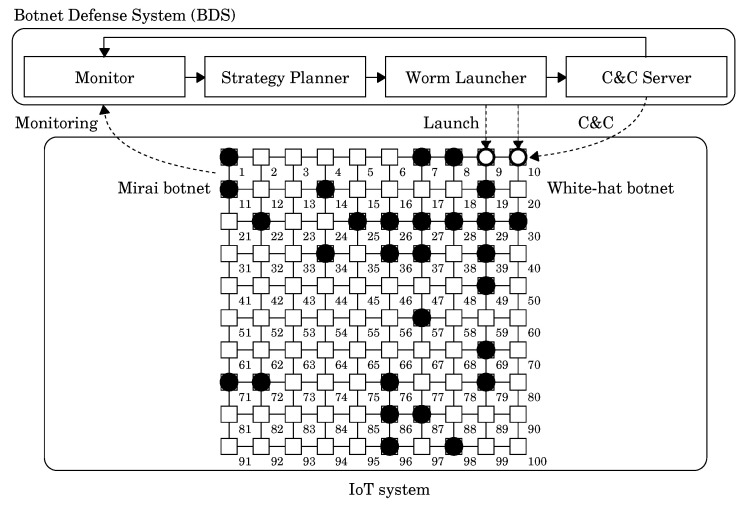
The overview of the Botnet Defense System.

**Figure 2 sensors-22-04666-f002:**
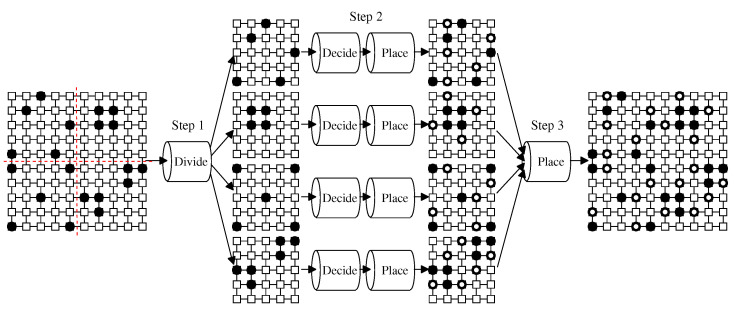
The operation of Procedure Worm-Launcher. After dividing a given 10 × 10 network into four 25 × 25 subnetworks, for each subnetwork, the Launcher predicts the placement of white-hat worms by machine learning models.

**Figure 3 sensors-22-04666-f003:**
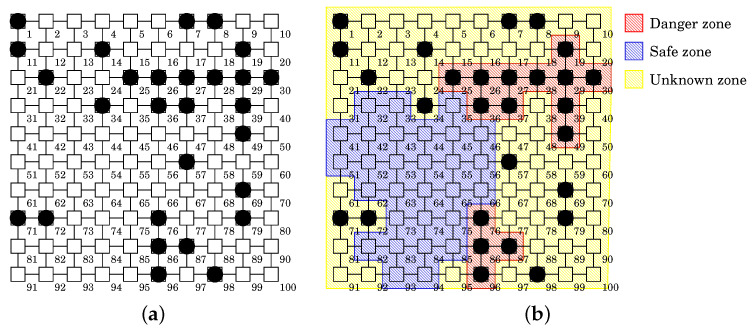
Application example of Procedure DBSCAN-based-Zoning. (**a**) Mirai bots’ distribution. (**b**) Result of zoning.

**Figure 4 sensors-22-04666-f004:**
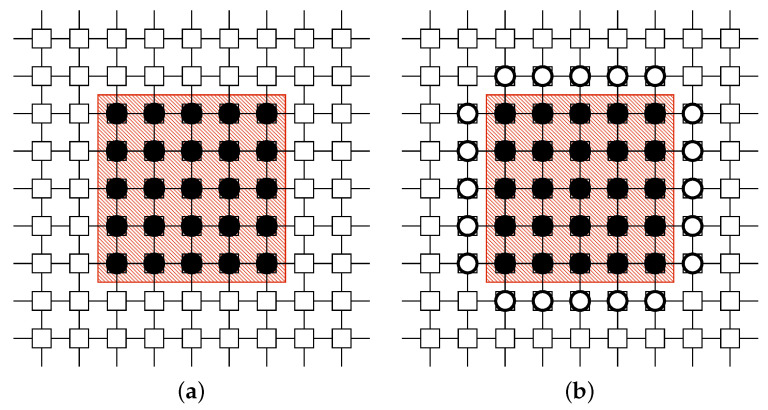
Application example of surrounding tactic for Danger Zone. (**a**) Before. The red area represents a Danger Zone. (**b**) After. The white-hat worms are placed to surround the zone.

**Figure 5 sensors-22-04666-f005:**
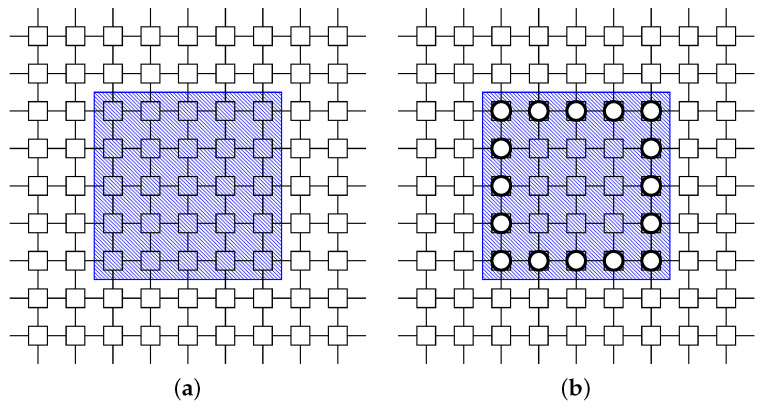
Application example of protecting tactic for Safe Zone. (**a**) Before. The blue area represents a Safe Zone. (**b**) After. The white-hat worms are placed to frame the zone.

**Figure 6 sensors-22-04666-f006:**
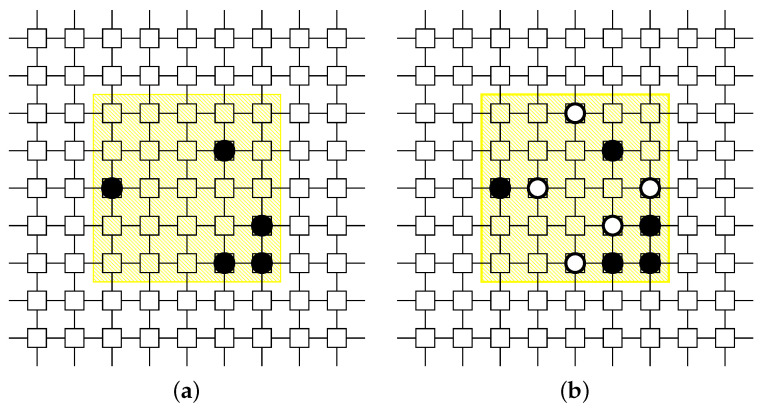
Application example of machine learning tactic for Unknown Zone. (**a**) Before. The yellow area represents an Unknown Zone. (**b**) After. The white-hat worms are placed according to the prediction of the given machine learning model.

**Figure 7 sensors-22-04666-f007:**
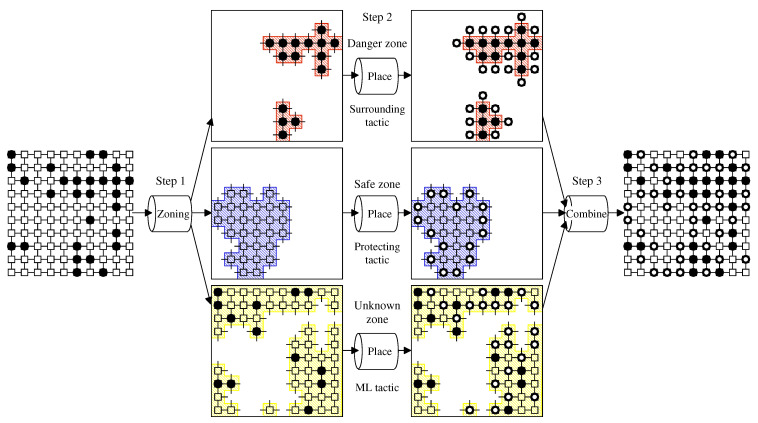
The operation of Procedure Zoning-based-Worm-Launcher. After dividing a given 10 × 10 network into three zones based on the Mirai bots’ density, the Launcher places white-hat worms according to the tactic for each zone.

**Figure 8 sensors-22-04666-f008:**
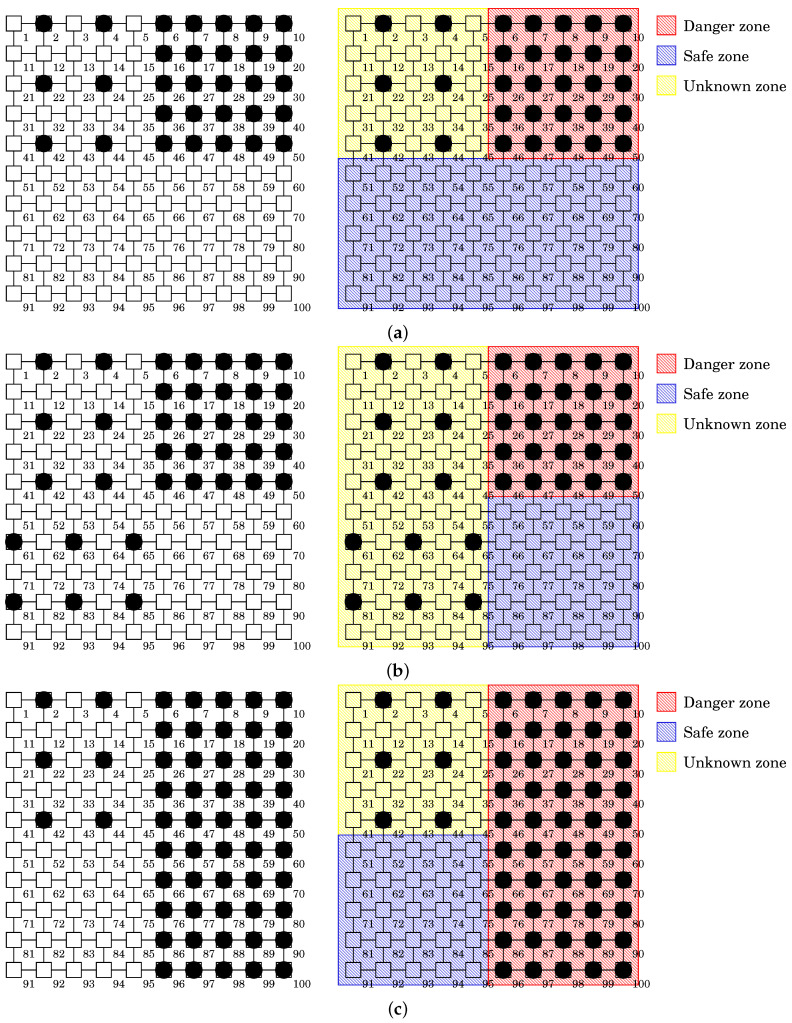
Three kinds of Mirai distributions in a 10 × 10 grid network used for simulation evaluation. (**a**) Distribution 1 and the result of zoning. (**b**) Distribution 2 and the result of zoning. (**c**) Distribution 3 and the result of zoning.

**Figure 9 sensors-22-04666-f009:**
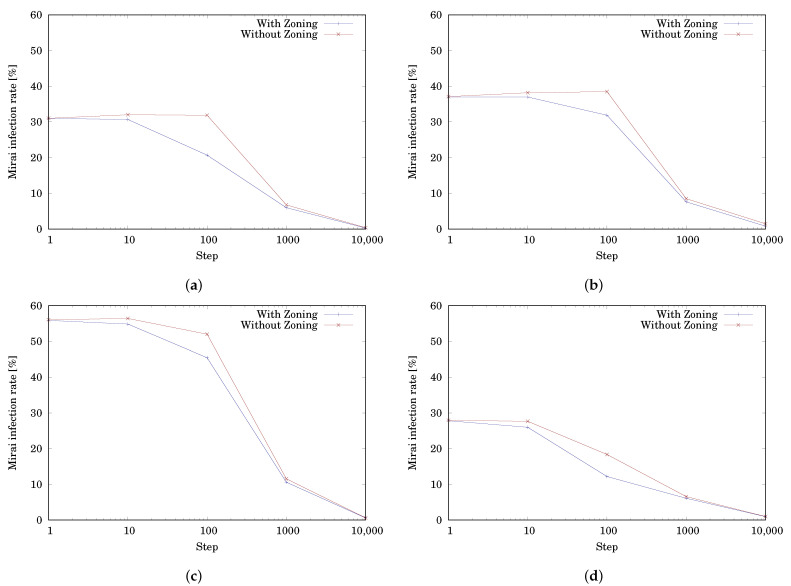
Comparison of the Launchers with and without zoning. (**a**) Distribution 1, (**b**) Distribution 2, (**c**) Distribution 3, and (**d**) Distribution 4.

**Table 1 sensors-22-04666-t001:** Overview of the detection methods.

Authors	Year	Attributes	Limitations	ML Algorithm ^1^	Eval. Measure
Qu et al. [17]	2018	Quantify the security risk in CPS network	No corresponding strategy for risk outbreak	N/A	Failure ratio
Nagisetty et al. [14]	2019	Monitor the network traffic by DL	Over-reliance on IoT traffic dataset	MLP, CNN	RMSE, F1-score
Vishwakarma et al. [20]	2019	Design a honeypot-based ML framework	Adaptability limitation to real-time scenarios	SVM, NB	N/A
Adan et al. [22]	2019	Design an IoT attack simulator	Limitation to public IoT networks	N/A	N/A
Soe et al. [15]	2020	Made sequential classifiers based on ML	Specific botnet can be classified	ANN, DT, NB	ACC
Guerra-Manzanares et al. [21]	2020	Generate traffic dataset of botnet	Low adaptability to real-time network	N/A	N/A
Wang et al. [19]	2020	Mine attack patterns from alarm logs	Lack of adaptability to local attack sequences	NB	ACC
Qu et al. [18]	2022	Detect attacks with AKF and GRU-CNN	Limitations for data loss and external interference	AFK, GRU-CNN	FPR, ACC
May raju et al. [16]	2022	Detect intrusion based on DRL	Compare with only two common ML models	DRL	ACC

^1^ MLP: multilayer perceptron, CNN: convolutional neural network, SVM: support vector machine, NB: naive Bayes, ANN: artificial neural network, DT: decision tree, AKF: adaptive Kalman filter, GRU: gated recurrent unit, DRL: deep reinforcement learning.

**Table 2 sensors-22-04666-t002:** Overview of the mitigation methods.

Authors	Year	Attributes	Limitations	Eval. Measure
Gopal et al. [27]	2018	Establish a security wall based on white listing	Lack of protection by other mechanisms	Router Model
Manso et al. [28]	2019	Construct a software-defined security system	Cause congestion of DNS controller	Quality of service
Ceron et al. [29]	2019	Modify the malicious traffic at network layer	Lack of dynamic malware analysis	Packet number

**Table 3 sensors-22-04666-t003:** Simulation result. Each value represents the mean of infection rate $R_{Mirai}$ for 1000 trials.

(a) Distribution 1		
Step	Without Zoning	With Zoning
1	37.11%	36.97%
10	38.20%	6.99%
100	38.50%	31.89%
1000	8.51%	7.62%
10,000	1.49%	0.82%
**(b) Distribution 2**		
Step	Without Zoning	With Zoning
1	31.03%	31.01%
10	32.03%	30.70%
100	31.89%	20.71%
1000	6.78%	5.99%
10,000	0.38%	0.29%
**(c) Distribution 3**		
Step	Without Zoning	With Zoning
1	0.0856.07%	0.0855.89%
0.0610	56.43%	54.86%
100	2.02%	45.47%
1000	11.56%	0.60%
10,000	0.62%	0.60%
**(d) Distribution 4**		
Step	Without Zoning	With Zoning
1	27.96%	27.82%
10	27.67%	25.99%
100	18.37%	12.20%
1000	6.58%	6.11%
10,000	1.00%	0.97%

**Table 4 sensors-22-04666-t004:** Positioning of this study among the related works.

Authors	Year	Detection	Mitigation	Extermination	Tactics	Zoning
Qu et al. [17]	2018	✓	×	×	×	×
Nagisetty et al. [14]	2019	✓	×	×	×	×
Vishwakarma et al. [20]	2019	✓	×	×	×	×
Adan et al. [22]	2019	✓	×	×	×	×
Soe et al. [15]	2020	✓	×	×	×	×
Guerra-Manzanares et al. [21]	2020	✓	×	×	×	×
Wang et al. [19]	2020	✓	×	×	×	×
Qu et al. [18]	2022	✓	×	×	×	×
May raju et al. [16]	2022	✓	×	×	×	×
Gopal et al. [27]	2018	×	✓	×	×	×
Manso et al. [28]	2019	✓	✓	×	×	×
Ceron et al. [29]	2019	×	✓	×	×	×
Yamaguchi [10]	2020	×	×	✓	×	×
Pan et al. [12]	2022	×	×	✓	✓	×
This study	2022	×	×	✓	✓	✓

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
