# Peer review of "Machine Learning White-Hat Worm Launcher for Tactical Response by Zoning in Botnet Defense System†"

_sensors, 2022, doi:10.3390/s22134666_

Round 1

Reviewer 1 Report

In this paper, the authors present a machine learning white-hat worm launcher for tactical response by zoning in botnet defense system. In general, this paper is well written and the topic is interesting. Here, there are some comments of this reviewer:

1 How scalable is the propose approach?

2 There is no discussion on the computational cost of the proposed method. What is the runtime? Please include such discussions.

3 The proposed method might be sensitive to the values of its main controlling parameters. How did you determine these parameters? Please elaborate on it.

4 Authors have not presented the limitations of this work. How this work can be extended in the future? 

5 The literature review can be strengthened. Authors are recommended to include and review the following study regarding application of machine learning for cyber security to improve the literature survey: “Active and passive hybrid detection method for power CPS false data injection attacks with improved AKF and GRU-CNN”, “Method for quantitative estimation of the risk propagation threshold in electric power CPS based on seepage probability”, "Method for extracting patterns of coordinated network attacks on electric power CPS based on temporal–topological correlation".

Author Response

Dear Reviewer,

We greatly appreciate your careful review.
We made the reply letter and revised the manuscript according to your indications.
Please find the attached file.

Best regards,
Shingo Yamaguchi

Reviewer 2 Report

Title: Machine Learning White-Hat Worm Launcher for Tactical Response by Zoning in Botnet Defense System

The research work is very effective and well presented, the proposed models produce very good results. However, there are some minor comments that need to be addressed.

·       Why the study has used only one clustering technique DBSCAN. Did they tried other clustering algorithms.

·       Because the DBScan has some drawback like it doesn’t work well with the varying density clusters and also the high dimensional clusters. how the authors addressed these limitations.

·       Table 1 should enlist the ML algorithms used in the previous studies. Furthermore, the results achieved by the previous studies interms of the performance evaluation measures.

·       The authors need to create a separate table for the detection and mitigation to provide the overview of the previous studies

·       The authors have compared the current proposed model with their previous two studies reference 10 and 12. The comparison also need to be extended to other studies in the literature that has used Mirai botnet.

·       Why Mirai botnet has been used in the study.

·       Overall the article is very well written.

Author Response

(The authors gave the same response as above.)

Round 2

Reviewer 1 Report

Thanks to the careful revision and detailed response made by the authors. All my concerns have been well addressed, and the revised manuscript has been much improved. I think this paper deserves to be published in its current form.